# Diversity of Cellular Slime Molds (Dictyostelids) in the Fanjing Mountain Nature Reserve and Geographical Distribution Comparisons with Other Representative Nature Reserves in Different Climate Zones of China

**DOI:** 10.3390/microorganisms12061061

**Published:** 2024-05-24

**Authors:** Zhaojuan Zhang, Meng Li, Shufei Zhang, Yue Qin, Jing Zhao, Yu Li, Steven L. Stephenson, Junzhi Qiu, Pu Liu

**Affiliations:** 1Engineering Research Center of Edible and Medicinal Fungi, Ministry of Education, Jilin Agricultural University, Changchun 130118, China; 17843098488@163.com (Z.Z.); limeng240312@126.com (M.L.); shufeizhang2004@outlook.com (S.Z.); 15843050165@139.com (Y.Q.); ytlyzjwjk1003@163.com (J.Z.); fungi966@126.com (Y.L.); 2Department of Biological Sciences, University of Arkansas, Fayetteville, AR 72701, USA; slsteph@uark.edu; 3Key Laboratory of Biopesticide and Chemical Biology, Ministry of Education, State Key Laboratory of Ecological Pest Control for Fujian and Taiwan Crops, College of Life Sciences, Fujian Agriculture and Forestry University, Fuzhou 350002, China

**Keywords:** dictyostelid diversity, geographical distribution, protected areas, soil protists, taxonomy

## Abstract

Protected areas are widely considered an essential strategy for biodiversity conservation. Dictyostelids are unique protists known to have important ecological functions in promoting soil and plant health through their top-down regulation of ecosystem processes, such as decomposition, that involve bacterial populations. But the relationship between dictyostelid diversity within protected areas remains poorly understood, especially on a large scale. Herein, we report data on the distribution of dictyostelids, identified with ITS + SSU rRNA molecular and morphology-based taxonomy, from soil samples collected in the Fanjing Mountain protected area of Guizhou Province, Southwest China. We compared the biodiversity data of dictyostelids in Fanjing Mountain with similar data from previously sampled sites in four other protected areas, including Changbai Mountain (CB), Gushan Mountain (GS), Baiyun Mountain (BY), and Qinghai–Tibet Plateau (QT) in China. We identified four species of dictyostelids belonging to three genera (*Dictyostelium*, *Heterostelium,* and *Polysphondylium*) and herein provide information on the taxonomy of these species. Two species (*Heterostelium pallidum* and *Dictyostelium purpureum*) are common and widely distributed throughout the world, but one species (*Polysphondylium fuscans*) was new to China. Our data indicate that there is no distinguishable significant correlation between the dictyostelid species studied and environmental factors. Overall, the similarity index between Baiyun Mountain in Henan Province and Fanjing Mountain in Guizhou Province, located at approximately the same longitude, is the highest, and the Jaccard similarity coefficients (Jaccard index) of family, genus, and species are 100%, 100%, and 12.5%, respectively. From a species perspective, species in the same climate zone are not closely related, but obvious geographical distributions are evident in different climate zones. This preliminary study provided evidence of the ecological adaptation of dictyostelids to different biological niches.

## 1. Introduction

Soil protists are ubiquitous yet remain largely unknown in terrestrial environments, although they have immense morphological and lifestyle diversity [1,2,3]. Dictyostelid social amoebae are highly diverse protists in soil, where they play an important role in bacterial population control and also the turnover of nutrients and minerals in nature [4,5]. They have a unique life cycle that begins with a unicellular (vegetative) stage, but when the food bacteria are consumed, the unicellular forms aggregate to form a multicellular stage in which fruiting bodies (sorocarps) [6,7] are formed (Figure 1A). Therefore, they offer a good system to study the evolution of multicellular complexity, with a well-resolved phylogeny and molecular genetic tools being available, among which *Dictyostelium discoideum* is now one of the most widely studied eukaryotic microbial model organisms [8,9,10].

Traditional morphology-based taxonomy of dictyostelids has been replaced by a molecular phylogeny. Their phylogenetic relationships were first reconstructed with molecular data based on SSU rDNA, which established four major groups of dictyostelids, simply named Groups 1, 2, 3, and 4 [11]. But now, the taxonomy of dictyostelids is based on the morphology of their fruiting body and a new classification with strong molecular phylogenetic support [12,13].

Protected areas have been considered a key part of an essential strategy for maintaining habitat integrity and species diversity [14,15], especially in the face of rapid urban expansion that has a profound impact on global biodiversity through habitat conversion, degradation, fragmentation, and species extinction [16,17]. However, the dictyostelids, as permanent inhabitants of the upper layers of soil, occupy a habitat that is suggestive of the primitive ecological environment [18]. Herein, we review the history of Chinese research on dictyostelids in light of the recent wave of research, with an increase in sample collection since 2010 (Figure 1B). There have been 58 species and varieties reported, which makes China one of the world’s biodiversity hotspots [19,20,21,22,23,24]. In fact, most of these species were from nature reserves. Guizhou Province is located in Southwest China, and has a wide range of soil types, vegetation, and climatic conditions along the gradient of elevation [25]. Previous studies of dictyostelid biodiversity have yielded only nine species (*Dictyostelium macrocephalum*, *D. implicatum*, *D. crassicaul, D. firmibasis, D. purpureum, Heterostelium tenuissimum, H. pallidum, Polysphondylium violaceum,* and *Cavenderia delicata*) from Guizhou Province [26,27].

Herein, using morphology and molecular systematics, we isolated and identified dictyostelids in the soil of the Fanjing Mountain Nature Reserve while also exploring their diversity and the impact of environmental factors on these species. More importantly, we analyzed the distribution characteristics of dictyostelids in current Chinese protected natural areas.

## 2. Materials and Methods

### 2.1. Study Area Description

The Fanjing Mountain Nature Reserve (27°49′50″ N to 28°1′30″ N and 108°45′55″ E to 108°48′30″ E) is located in the central portion of the Tongren region in the northeast of Guizhou Province, China, with an elevation of 2572 m and a total area of 567 km^2^ (Figure 2A). The Fanjing Mountain Nature Reserve is in the subtropical humid monsoon climate zone, with an average temperature of 13.1–14.7 ℃. The frost-free period is 270–278 days, and the average annual precipitation is 1100–2600 mm [28,29].

### 2.2. Experimental Design and Sampling

This survey was conducted in late August 2022, which falls within the midsummer season that occurs throughout the year. During this period, the topsoil has completely thawed and the vegetation and biomass of the forest are at their peak. The reserve has the largest distribution area of yellow mountain soil and dark yellow-brown soil, which are mainly forest soils and largely undisturbed by human activities.

We set up a single plot for each forest type (coniferous forest = P1, broadleaf forest = P2, mixed forest = P3) in the study areas, and the plot was randomly chosen on a 300 × 300 m grid (Figure 2B) with respect to three representative forest types (Figure 2C–F) that were selected, with three replicates (2 × 2 m quadrats) prepared for each plot. To avoid the effects of distance on the microbial community, the interval between each quadrat was controlled to be less than 100 m, and nine points in each quadrat were randomly selected for sampling. 

In each quadrat, we used sterile drills to collect soil after the litter layer above the soil had been removed. Afterwards, the mixed samples from nine sampling points were combined into one sample, and about 100 g of this was placed into disposable sterile ziplock bags, marked with number, latitude and longitude, elevation, vegetation type, air humidity, temperature, and weather conditions (Table 1). All these samples were numbered and recorded in the laboratory soil sample database and then stored at 4 °C in a refrigerator.

### 2.3. Sample Processing

**(1) Dictyostelid isolation.** The isolation methods used in the present study followed those described by Cavender and Raper [30]. A portion (10 g) of each soil sample was mixed with 90 mL of distilled water and shaken at 280 rpm/min at 23 ℃ for 2 min to prepare an initial soil suspension of 1:10. Soil water moisture was determined using gravimetric analysis [31]. The pH values of the soil dilutions were determined with a PHS-3C pH meter. Afterwards, 5 mL of the soil suspension was mixed with 7.5 mL of sterile water to make a soil suspension of 1:25. Then, 0.5 mL of the soil suspension and 0.4 mL of a suspension of *Escherichia coli* (food) were uniformly spread over the surface of a hay infusion agar medium using an applicator. 

Each soil sample was set in up in six parallels; each plate contained 0.02 g soil, and the total soil of the six plates was 0.12 g. Each inoculated plate was examined carefully at least once a day following the appearance of initial aggregations of amoebae or fruiting bodies that had developed from the propagules present in the sample material, and each aggregation/fruiting (“clone” [30]) was marked (Figure 3A). The formulae used to evaluate the data were calculated as follows [30,32]:(1)$$\mathrm{Density}\;(\mathrm{clones}/g)=\frac{\mathrm{Number}\;\mathrm{of}\;\mathrm{dictyostelid}\;\mathrm{clones}}{\mathrm{Dry}\;\mathrm{soil}\;(g)}$$
(2)$$\mathrm{Frequency}\;(\%)=\frac{\mathrm{Number}\;\mathrm{of}\;\mathrm{dictyostelid}\;\mathrm{occurrences}}{\mathrm{Quadrats}}$$
(3)$$\mathrm{Relative}\;\mathrm{density}\;(\%)=\frac{\mathrm{Clones}\;\mathrm{of}\;\mathrm{each}\;\mathrm{dictyostelid}\;\mathrm{species}}{\mathrm{Total}\;\mathrm{number}\;\mathrm{of}\;\mathrm{clones}}$$
Jaccard similarity coefficient (S_J_) (%) = [a/(a + b + c)] × 100%(4)
where S_J_ is the similarity coefficient of the corresponding classification units in the two comparison areas, a is the number of shared classification units, b and c are the numbers of classification units that only appear in one place, respectively.

**(2) Dictyostelid purification.** This was followed by adding 0.4 mL of a suspension of *E. coli*, which was spread evenly over the surface of the agar. Five duplicate plates were prepared for each sample and placed in an incubator at 23 °C with a 12 h light and dark cycle. After two to three days, the plates were examined and any dictyostelids observed were recorded. The isolates appearing in the plates were purified and cultivated for taxonomic studies on non-nutrient water agar plates with *E. coli* pregrown for 12 to 24 h. 

**(3) Dictyostelid preservation.** Spores from these isolates were frozen in HL5 [33] media and stored at −80 °C in the Herbarium of the Mycological Institute of Jilin Agricultural University (HMJAU), Changchun, China.

### 2.4. Morphological Observations

In the non-nutrient water agar plates, the location of each early sorocarp(s) that developed was marked. The characteristic stages in the life cycle, including cell aggregation and the formation of pseudoplasmodia and sorocarps, were observed under a Zeiss dissecting microscope (Axio Zoom V16; Zeiss, Oberkochen, Germany) with a 1.5× objective and 10× ocular. Slides with sorocarps were prepared with water as the mounting medium. Features of spores, sorophores, and sorocarps were observed and measured on the slides by using a Zeiss light microscope (Axio Imager A2; Zeiss, Oberkochen, Germany), with a 10× ocular and 10, 40, and 100× (oil) objectives (Figure 3B). Photographs were obtained with a (Zeiss Axiocam 506; Zeiss, Oberkochen, Germany) color microscope camera.

### 2.5. DNA Isolation, PCR Amplification, Sequencing, and Phylogenetic Analysis

The molecular methods used in the present study followed those described by Liu et al. [22]. The genomic DNA solution was used directly for the PCR amplification using the primers 18SF-A (5′AACCTGGTTGATCCTGCCAG3′), 18SR-B (5′TGATCCTTCTGCAGGTTCAC3′) [34] and ITS1 (5′TCCGTAGGTGAACCTGCGG3′), ITS4 (5′TCCTCCGCTTATTGATATGC3′) [35]. PCR products were sent to Sangon Biotech Co., Ltd. (Shanghai, China), for sequencing (Figure 3C). Sequences obtained were deposited in the GenBank database. The eight newly generated sequences were checked and then submitted to GenBank. All the sequences of closely related species were downloaded from GenBank for phylogenetic analysis to determine their phylogenetic relationships with other taxa in the group (Appendix A). The ITS and SSU sequences were aligned and compared separately by using the program MAFFT v7.505 [36], then manually adjusted in BioEdit version 7.0.9.0 [37]. Maximum-likelihood analyses (ML analyses) were performed using IQTREE v.1.6.12 [38], with 10,000 replicates of ultrafast-likelihood bootstrapping to obtain node support values with the “-bb 10,000” option, and further optimized using a hill-climbing nearest-neighbor interchange (NNI) with the “-bnni” option [39]. We also used the SH-aLRT test to obtain the confidence limit of the topology with the “-alrt 1000” option [40]. The “-nt AUTO” option was used to automatically determine the best number of cores given the current data. We used ModelFinder as implemented within IQ-TREE to determine the best substitution model based on Bayesian information criteria (BIC) [41]. We used a myxomycete sequence (Physarum polycephalum, accession no. X13160.1/NC_002508.1) as the outgroup with the “-o X13160.1”/ “-o NC_002508.1” option.

### 2.6. Data Collection of Dictyostelids in Four Other Nature Reserves

In order to clarify the geographical distribution of dictyostelids in the nature reserves, four other representative protected areas in China were selected along with Fanjing Mountain Nature Reserve in this study; the data sources for the four representative protected areas included seven species from Changbai Mountain (CB) [23] (*Dictyostelium discoideum*, *D. mucoroides*, *D. robusticaule*, *Cavenderia fasciculata*, *Heterostelium pallidum*, *H. recretum*, *H. candidum*), three species from Gushan Mountain (GS) [42] (*Dictyostelium brefeldianum*, *D. clavatum*, *D. purpureum*), five species from Baiyun Mountain (BY) [43] (*Dictyostelium sphaerocephalum*, *D. purpureum*, *Heterostelium candidum*, *H. tenuissium*, *Polysphondylium violaceum*), and 12 species from the Qinghai–Tibet Plateau (QT) [22] (*Dictyostelium brevicaule*, *D. vermiforme*, *D. brefeldianum*, *D. sphaerocephalum*, *D. minimum*, *D. crassicaule*, *D. multiforme*, *Cavenderia fasciculata*, *C. antarctica*, *C. aureostipes*, *C. exigua*, *Heterostelium tikalense)*.

### 2.7. Statistical Analyses

#### 2.7.1. Dictyostelid Diversity Analyses

All dictyostelid diversity analyses were generated using the Hiplot Pro Biomedical Visualization Platform (available online: https://hiplot.com.cn/home/index.html, accessed on 12 April 2023). The data on community calculation—density (clones/g), frequency (%), and relative density (%)—were analyzed based on the Excel table. The DotChart Plot (Figure 4A) of dictyostelid clones from all soil samples was used to evaluate dictyostelid density in three plots; the Pyramid Chart (Figure 4B) of relative abundance and frequency of dictyostelid species was used to evaluate the dominant species.

#### 2.7.2. Environmental Impacts on Dictyostelid Communities

The influence of environmental factors on the dictyostelid communities of the collecting locality was analyzed by redundancy analysis (RDA, which is based on a linear model) (Section 3.4) using the cloud platform (available online: http://www.cloud.biomicroclass.com/, accessed on 18 April 2023). First, we sorted out the species abundance table, the list of environmental factors, and the grouping files in Excel. Then, these three files were respectively converted to the “txt” format. Environmental factors include edaphic factors (e.g., water moisture, pH), biotic factors (e.g., vegetation), spatial factors (e.g., elevation, longitude, and latitude), and climatic factors (e.g., temperature, air humidity).

#### 2.7.3. Geographic Distribution of Dictyostelids in Nature Reserves

In order to compare the relevance of dictyostelid species in the nature reserves, the Jaccard similarity coefficient (S_J_) was applied to calculate the value of similarity between species classification units (family, genus, species) (Section 3.5), the cloud platform (http://www.cloud.biomicroclass.com/, accessed on 23 April 2023) was used to perform a principal component analysis (PCA) (Section 3.5), and the Stacked Column Chart with a clustering tree (Section 3.5) based on Bray–Curtis distances was used to detect the geographical distribution of dictyostelids in the nature reserves.

## 3. Results

### 3.1. Dictyostelid Community Composition and Diversity

Nine mixed soil samples from three plots in the Fanjing Mountain Nature Reserve were screened for clones using pure cultivation. A total of 34 clones were isolated from the soil samples in this study; only in a single plot (plot 3) did the number of clones recorded exceed 30, and in two others the number of clones was less than five (Table 1). Moreover, the number of clones in quadrat 3 was 31, accounting for 91% of the total number of clones (Table 1). Numbers of clones per gram of sample material varied widely, and the highest density for a particular locality (86.11 clones/g) was recorded for a mixed forest, whereas the lowest density (2.78 clones/g) was obtained for samples from a coniferous forest in the Fanjing Mountain Nature Reserve (Figure 4A). *Heterostelium pallidum* was the most widespread species and was recorded in mixed forests of quadrat C2 and quadrat C3 in the Fanjing Mountain Nature Reserve, which had the highest relative density and frequency (Figure 4B). 

### 3.2. Dictyostelid Species

A total of four species were obtained from nine quadrats in three plots from which soil samples were collected in the Fanjing Mountain Nature Reserve, Guizhou Province, during August 2022. Among these, the species recovered were classified as follows: *Polysphondylium fuscans* (clone no. = 1), *Dictyostelium purpureum* (clone no. = 1), *D. robusticaule* (clone no. = 2), and *Heterostelium pallidum* (clone no. = 30) (Table 1). These four species recorded in the entire study were present in only four quadrats. In contrast, no isolates were recovered from the other five quadrats processed. Soil samples from quadrat C2 of the Fanjing Mountain Nature Reserve (collected from a mixed forest at 27°50′39″ N, 108°47′13″ E) yielded the largest number of species, a total of two species. *P. fuscans* (Figure 5) and *D. robusticaule* (Figure 6) were not previously known from Guizhou Province, and *P. fuscans* is a new record for China. Two species were *D. purpureum* (Figure 7) and *H. pallidum* (Figure 8), which were common in Guizhou Province. The morphological characteristics of these species vary widely in shape, size, and coloration among species, and most of the characteristics used for species identification are described below (Table 2).

### 3.3. Complete Sequencing and Phylogenetic Analysis

Among the eight new SSU rDNA and ITS rDNA sequences obtained in this study, the lengths of these sequences in dictyostelids ranged from 1600 to 1800 bp (SSU) and 700 to 900 bp (ITS), respectively. They shared 99% to 100% similarity with each other and 99% to 100% similarity with the closest strain in GenBank. A maximum-likelihood phylogenetic tree of the complete SSUrDNA sequences was constructed based on the strains from this study as well as all other known dictyostelids (Figure 9A). A maximum-likelihood phylogenetic tree of the partial ITS rDNA sequences was constructed based on the strains from this study as well as representative strains from GenBank with high similarity to our strains (Figure 9B).

The phylogenetic studies of the SSU rDNA data revealed the four species are members of the orders Dictyosteliales and Acytosteliales, which can be divided into 3 distinct branches within 12 genus branches. Two strains (HMJAU C211 and HMJAU B341) belong to *Dictyostelium* (sorocarps unbranched or irregular branches, spores without polar granules and polar granules unconsolidated) and cluster together with high support, and are closely related to *Dictyostelium purpureum* QSpu36 (GenBank accession no. FJ424828.1) and *Dictyostelium robusticaule* 5729-bai-2021 (GenBank accession no. MW931857.1), respectively. The SH-aLRT support/ultrafast bootstrap support values = 0/80 and 76.7/97 (Figure 10A). One strain (HMJAU A241) belongs to *Polysphondylium* (sorocarps with violet/lavender/purple sori, branches irregular or regularly spaced whorls, spores with consolidated polar granules) and cluster together with high support, which indicates that they are closely related to *Polysphondylium fuscans* Sweden-11D (GenBank accession no. JX173877.1), with an SH-aLRT support/ultrafast bootstrap support value of 87.9/91 (Figure 10B). One strain (HMJAU C345) belongs to *Heterostelium* (sorocarps sparsely branched, spores with granules present/absent, consolidated/unconsolidated) and clusters together with high support, which indicates a close relationship to *Heterostelium pallidum* TNS-C-98 (GenBank accession no. AM168103.1 and an SH-aLRT support/ultrafast bootstrap value of 98.8/83, Figure 11).

### 3.4. Environmental Responses of Dictyostelid Communities in Different Habitats

Redundancy analysis was applied to determine correlations between eight environmental factors in four categories and dictyostelid communities. The RDA results showed that environmental factors explained 99.3% and 0.47% of the variation in soil dictyostelid communities among the environmental factors in the Fanjing Mountain Nature Reserve in this study. Among the environmental data, the environmental factors had no distinguishable significant correlation with variations in dictyostelid communities (*p* > 0.05, Figure 12), as they accounted for 20.51% and 36.54% of the total variation, respectively.

### 3.5. Comparison of Similarity and Geographical Distribution of Dictyostelids in Representative Natural Reserves

The Jaccard similarity coefficient (S_J_ %) of the family, genus, and species was calculated using data obtained from Changbai Mountain (CB) [23] in Jilin Province, Gushan Mountain (GS) [42] in Fujian Province, Baiyun Mountain (BY) [43] in Henan Province, Qinghai–Tibet Plateau in Tibet (QT) [22], and the Fanjing Mountain Nature Reserve (FJ) in Guizhou in the present study (Figure 13A). At the family level, the majority for each other (S_J_ %) was higher than 50%, reaching moderate similarity. At the family level, the values (S_J_ %) for FJ and BY, CB, and QT were 100%. A total of only one family of the Dictyosteliaceae was the same in five reserves. At the genus level, the comparison was generally similar. The difference was mainly manifested in the abundance of dictyostelids in CB and QT, where dictyostelid distribution was obviously unbranched or had irregular branches, including *Dictyostelium*, *Heterostelium*, and *Cavenderia*. At the species level, FJ and CB had the highest species similarity coefficient (S_J_ %), which was 22.2%, while others were 5.56%–16.7%. Although the family similarity coefficient (S_J_ %) was relatively high, the interspecies similarity values were all low in these five reserves. We found that the dictyostelids have different environmental requirements, which also reflects their unique ecological adaptability.

Using the dictyostelid species from those five protected areas noted above, the distribution of dictyostelids in three climate zone protected areas were elucidated using principal component analysis (PCA) and a clustered bar chart (Figure 13B,C). Dictyostelids were mainly assigned to subtropical (FJ and GS), temperate (CB and BY), and plateau (QT) climatic zones, with proportions of 40%, 40%, and 20%, respectively. The PCA results demonstrated that resultant principal components (PCs) explained 90.47% of the variance. There was some similarity between the dictyostelids (FJ) and the dictyostelids (GS) in the subtropical climatic zone. However, the most distinct differences in community composition were observed between the dictyostelids (CB) and dictyostelids (BY) in the temperate climatic zone. In addition, the dictyostelid (QT) community structure in the plateau climate zone was quite distinct from those of both the subtropical and the temperate climatic zones (Figure 13B). It is worth noting that *D. purpureum* appeared in all three (FJ, GS, BY) protected areas (Figure 13D).

## 4. Discussion

### 4.1. Taxonomy and Molecular Phylogeny

In this study, we isolated 34 strains which could be assigned to four species based on morphology and as confirmed by molecular sequence data (Table 2; Figure 9, Figure 10 and Figure 11). They belong to two families in the Dictyosteliaceae and Acytosteliaceae, with three genera (*Dictyostelium*, *Heterostelium*, and *Polysphondylium*) from Guizhou Province, Southwest China, as determined by laboratory cultivation (Figure 2; Table 1). Among these, *P. fuscans* was new to China; its sori (Figure 5A,B) start out colorless and develop violet coloration that darkens with age. This places the species in a highly supported clade together with *P. fuscans* (GenBank accession no. JX173877.1) in an SSU rRNA phylogeny (Figure 10B). Morphologically, *P. fuscans* is most similar to *Polysphondylium violaceum* but differs primarily in the lighter initial pigmentation of the sori, the smaller number of whorls, and the larger spore size. This species was first reported from mildly acidic soils (pH = 5.5) in a coniferous forest in central Sweden [44]. Surprisingly, the species isolated in this study also came from coniferous forests with mildly acidic soils (pH = 5.7), indicating the species’ preference for this type of soil habitat. *Dictyostelium purpureum* (Figure 7), with large, robust sorocarps and deep purplish sori, is a cosmopolitan species reported from Africa, across North America, and in many localities in Asia [45]. *Dictyostelium robusticaule*, with the species name referring to its robust sorophores, was first reported by Zou et al. [23] from a mixed forest on Changbai Mountain, China. In this study, the species was isolated from a broadleaf forest, and SSU rRNA phylogeny places it in a highly supported clade together with *D. robusticaule* (GenBank accession no. MW931857.1) (Figure 10A). *Heterostelium pallidum* is characterized by whirls of sori, which places this species in a highly supported clade together with *H. pallidum* (GenBank accession no. AM168103.1) in an SSU rRNA phylogeny (Figure 11). Stephenson and Landolt [46] studied canopy soil in Puerto Rico and showed that *H. pallidum* tends to be found in regions with a tropical or subtropical climate. Furthermore, Cavender [47] suggested that *H. pallidum* is one of the few cellular slime molds that seem to be indifferent to climatic conditions. 

### 4.2. Environmental Factors Drive the Effects of Dictyostelid Diversity 

At present, numerous regions in China (Figure 1A) have carried out extensive sampling to investigate dictyostelids, with identification mostly based on purely descriptive species information. However, the ecology of these extremely unique populations is still poorly understood, and ecological factors are rarely associated with the diversity distribution of this group’s organisms. Previously, some researchers reported ecological information from China, North America, Central America, and East Africa [24,48,49,50,51] along with Madagascar [50] and Christmas Island [52], thus providing information that deepened our interest in this particular group.

In general, the overall diversity and abundance of dictyostelids in the Fanjing Mountain Nature Reserve appear to be low. As noted above, five (A1, A3, B1, B2, and C1) of the nine quadrats yielded no clones, and a total of only 34 clones were recovered from four samples (Table 1). Many factors affect the distribution and density of dictyostelids, such as abiotic factors (moisture, organics, pH, temperature, soil quality, and elevation) and biotic factors (prey–predator and other interspecies interactions) [53,54]. The growth and reproduction of soil protozoa largely rely on the habitat provided by vegetation, which not only changes the local microclimate but affects the nutrients, pH, water content, oxygen content, and temperature through litter or root exudates [55,56]. In this study, the environmental factors had no distinguishable significant correlation with variations in dictyostelid communities (*p* > 0.05, Figure 12). Our results found that the mixed forest with a total of 31 clones isolated had the highest density (86.11 clones/g), including two species, *D. purpureum* (clone no. = 1) and *H. pallidum* (clone no. = 30) (Table 1), which is consistent with a previous report indicating that the highest species abundance occurs in mixed forests in Changbai Mountain [23]. The reason is that through complementary interactions, mixed forests are more resistant to relatively small-scale and selective natural disturbances; therefore, productivity may be higher than in a single forest type [57,58].

### 4.3. Comparative Analysis of Geographical Distribution and Species Similarity in Natural Reserves

All the current research on protected areas in China summarized above makes the facilitation of cooperation among scientists from different countries and continents particularly important. For dictyostelid distribution, the similarity between the families and genera of dictyostelids in different regions ranges from 33.3% to 100%; the similarity of species ranges from 0 to 22.2% (Figure 13A). Based on the available information on types of dictyostelids and the literature records, the geographical distribution of dictyostelids in five protected areas was divided into three types: subtropical distribution (6 species), temperate distribution (11 species), and plateau distribution (12 species). The proportions of distributional types were 26.09%, 47.83%, and 52.17%, respectively (Figure 13C). The frequency of *D. purpureum* was the highest, with it occurring in three (FJ, GS, BY) protected areas. Swanson et al. [54] reported earlier on the known distributional data of dictyostelids found in forest soils around the world, based on data available at that time. The distribution of these species can be divided into four categories: cosmopolitan, intermittent, limited, and subtropical. Drawing inspiration from these distribution categories, it seems that in addition to the cosmopolitan species (e.g., *D. purpureum* and *H. pallidum*), other species (e.g., *P. fuscans* and *D. minimum*) in different regions have certain geographical limits based on the overall data from this study.

In addition, from an overall perspective, the species distribution in the subtropical climatic zone (FJ and GS) is characterized by small spatial distances, while that in the temperate climatic zone (CB and BY) is more distant. Of course, this is also influenced by the number of samples and various environmental factors [47], so more researchers need to be involved in the discovery of species. This indicates that more repetitive samplings across broader temporal and spatial scales are needed to accurately describe the true geographical distribution of this group of organisms, based on big data and interdisciplinary approaches.

## 5. Conclusions

In this study, 34 isolates representing four species of dictyostelids were obtained from soil samples collected from Fanjing Mountain in Guizhou Province, China. One species (*Polysphondylium fuscans*) was new to China, which provides a new starting point for the study of this species. Our data indicate that the distribution patterns of dictyostelids are most influenced by vegetation and temperature, which provides important information for understanding the ecological characteristics of these organisms. These obvious geographical distribution patterns of dictyostelids in different climate zones have great significance for developing a better understanding of their ecological adaptability and evolutionary history.

However, in future studies, we should (1) consider the limitations of the research area, such as by increasing the collection of samples and adopting new technologies such as high-throughput sequencing; (2) consider the complexity of environmental factors, such as soil conditions and the influence of biological factors; and (3) propose specific protection measures or recommendations to help protect these precious biological resources.

## Figures and Tables

**Figure 1 microorganisms-12-01061-f001:**
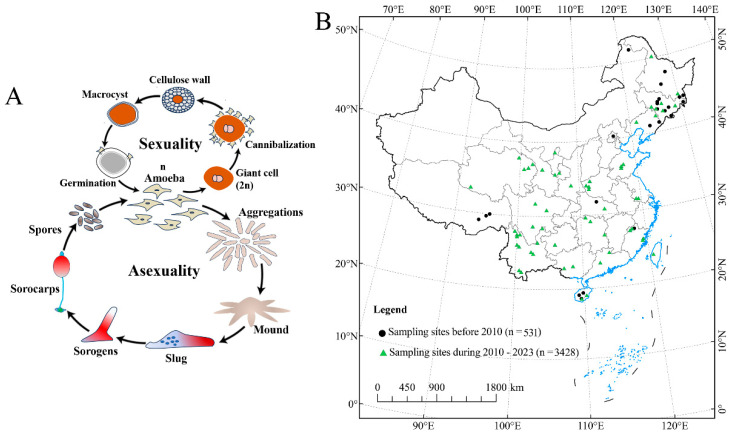
Changes in the distribution of sampling sites used to assess dictyostelid communities in China and the life cycles of Amoebozoan groups related to the dictyostelids. (**A**) The dictyostelids show both asexual and sexual cycles. (**B**) The legend of “sampling sites” is based on a literature analysis (i.e., China National Knowledge Infrastructure, CNKI).

**Figure 2 microorganisms-12-01061-f002:**
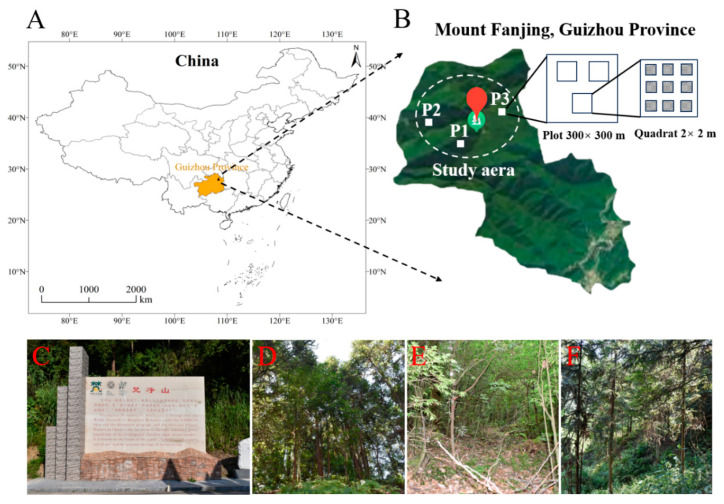
Location of the Fanjing Mountain Nature Reserve in China. (**A**) Map of the Fanjing Mountain Nature Reserve in the northeast of Guizhou Province, China. (**B**) Study area (nine-point sampling method) used in the Fanjing Mountain Nature Reserve. (**C**) Image of the Fanjing Mountain Nature Reserve. (**D**) Mixed forest. (**E**) Broadleaf forest. (**F**) Coniferous forest.

**Figure 3 microorganisms-12-01061-f003:**
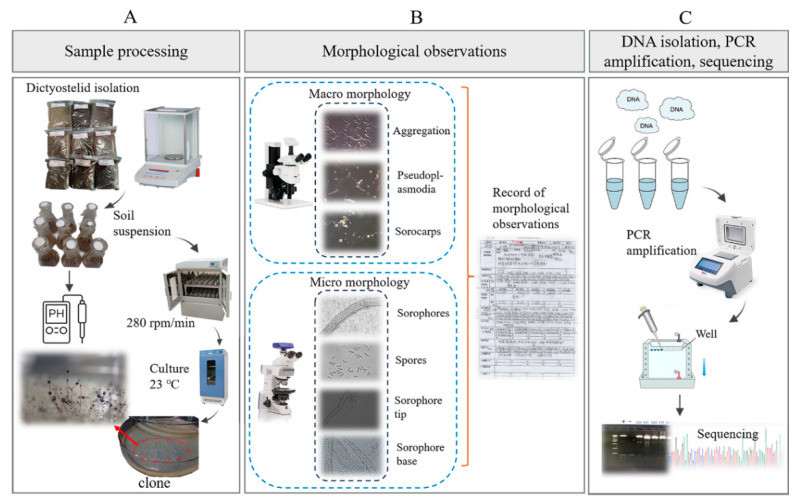
Outline of the experimental workflow used to isolate dictyostelids from the collected soil samples. (**A**) Sample processing. (**B**) Morphological observations. (**C**) DNA isolation, PCR amplification, sequencing.

**Figure 4 microorganisms-12-01061-f004:**
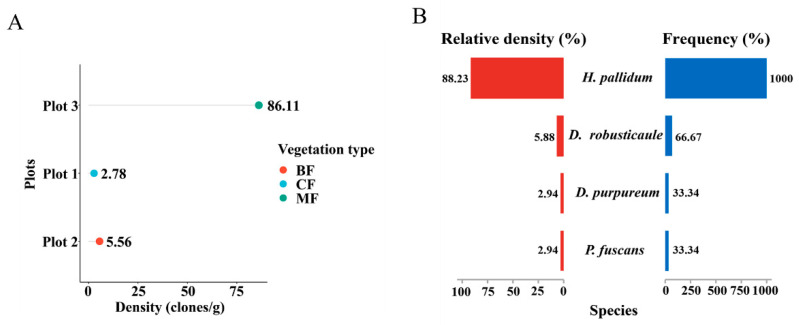
Comparison of the community diversity of dictyostelids along the different dimensional diversity levels. (**A**) Density in the plots. Abbreviations: broadleaf forest (BF), coniferous forest (CF), mixed forest (MF). (**B**) The relative abundance and frequency of species.

**Figure 5 microorganisms-12-01061-f005:**
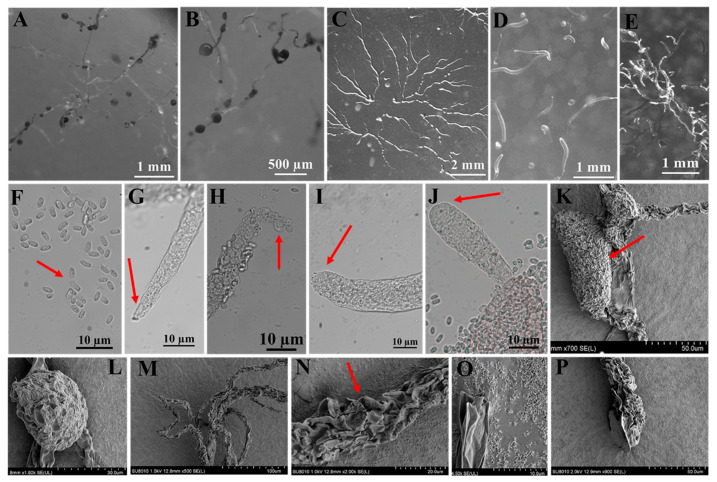
Morphological features of *Polysphondylium fuscans* Perrigo and Romeralo (HMJAU A241). (**A**,**B**) Sorocarps. (**C**) Aggregations. (**D**) Pseudoplasmodia. (**E**) Clustered sorogens. (**F**) Spores. (**G**,**H**) Sorophore tips. (**I**,**J**) Sorophore bases. (**K**,**L**) Sorus (SEM). (**M**–**O**) Sorophores (SEM). (**P**) Sorophore base (SEM). The red arrows refer to the key distinguishing characteristics.

**Figure 6 microorganisms-12-01061-f006:**
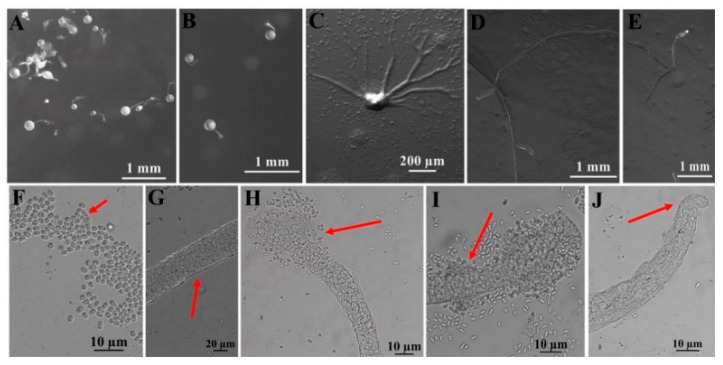
Morphological features of *Dictyostelium robusticaule* Y. Li, P. Liu, Y. Zou (HMJAU B341, B321). (**A**,**B**) Sorocarps. (**C**) Aggregations. (**D**,**E**) Pseudoplasmodia. (**F**) Spores. (**G**) Sorophore. (**H**,**I**) Sorophore tips. (**J**) Sorophore base. The red arrows refer to the key distinguishing characteristics.

**Figure 7 microorganisms-12-01061-f007:**
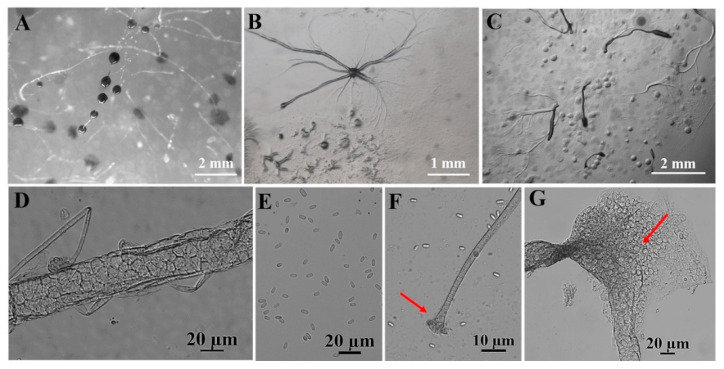
Morphological features of *Dictyostelium purpureum* Olive (HMJAU C211). (**A**) Sorocarps. (**B**) Aggregations. (**C**) Pseudoplasmodia. (**D**) Sorophore. (**E**) Spores. (**F**) Sorophore tip. (**G**) Sorophore base. The red arrows refer to the key distinguishing characteristics.

**Figure 8 microorganisms-12-01061-f008:**
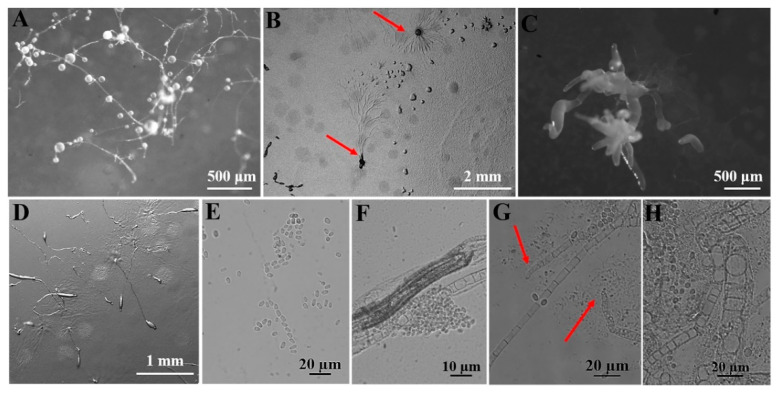
Morphological features of *Heterostelium pallidum* (Olive) S. Baldauf, S. Sheikh, and Thulin (HMJAU C231; C311-C315; C321-C324; C331-C336; C341-C347; C351-C357). (**A**) Sorocarps. (**B**) Aggregations. (**C**) Clustered sorogens. (**D**) Pseudoplasmodia. (**E**) Spores. (**F**) Sorophores. (**G**) Sorophore tips. (**H**) Sorophore base. The red arrows refer to the key distinguishing characteristics.

**Figure 9 microorganisms-12-01061-f009:**
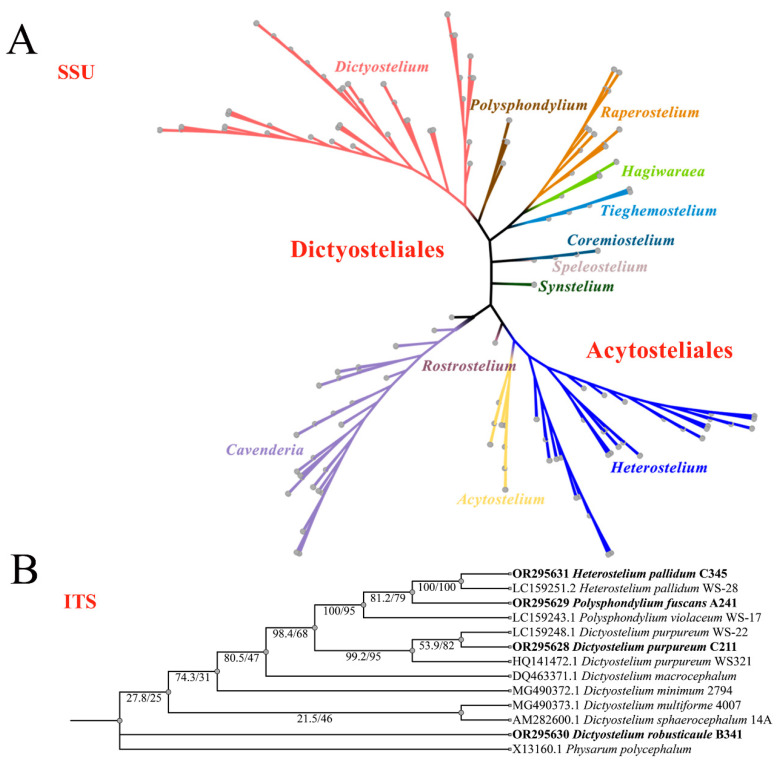
Phylogeny of all known dictyostelids based on SSU rDNA (**A**) and closely related species of dictyostelids based on ITS rDNA, newly generated sequences are indicated in bold (**B**).

**Figure 10 microorganisms-12-01061-f010:**
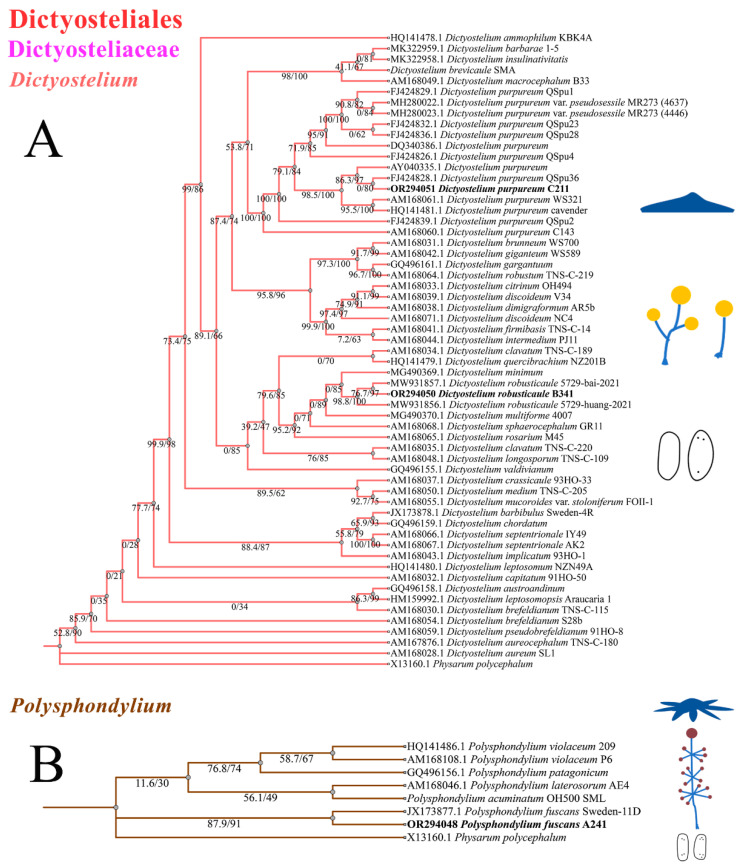
SSU phylogeny of *Dictyostelium* (**A**) and *Polysphondylium* (**B**) sequences in the order Dictyosteliales and the family Dictyosteliaceae. Numbers in parentheses are SH-aLRT support (%)/ultrafast bootstrap support (%). Newly generated sequences are indicated in bold.

**Figure 11 microorganisms-12-01061-f011:**
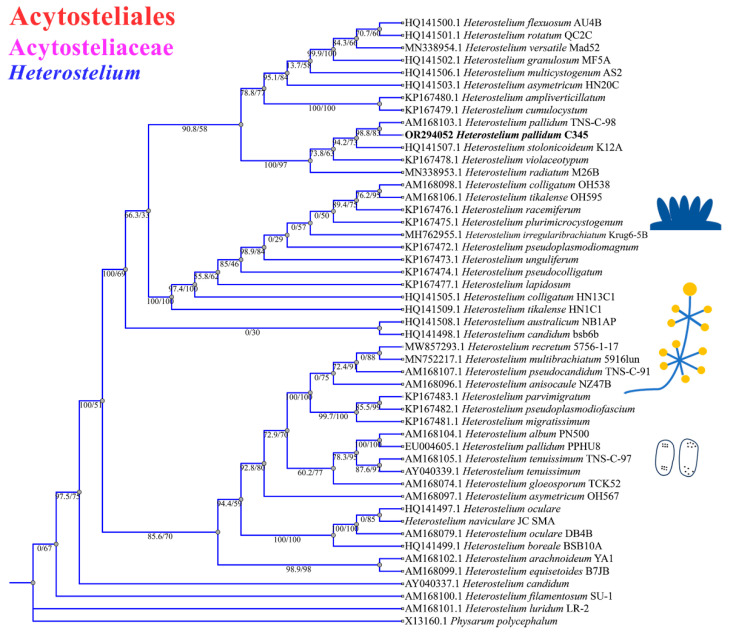
SSU phylogeny of *Heterostelium* sequences in the order Acytosteliales, family Acytosteliaceae. Numbers in parentheses are SH-aLRT support (%)/ultrafast bootstrap support (%). Newly generated sequences are indicated in bold.

**Figure 12 microorganisms-12-01061-f012:**
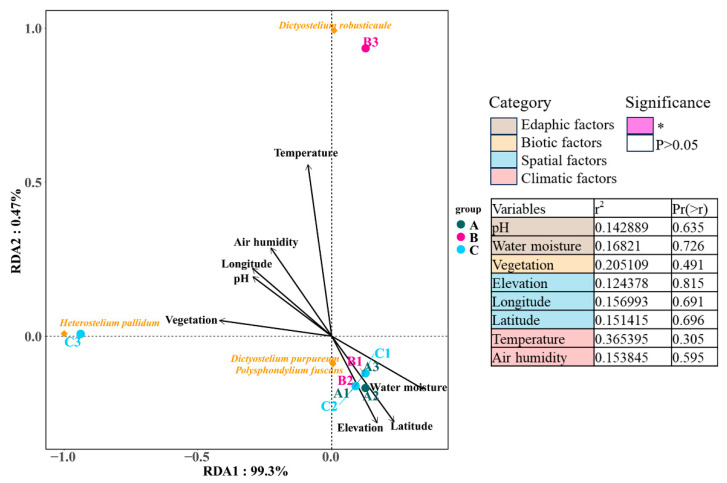
The edaphic, biotic, spatial, and climatic factors explaining the diversity of dictyostelid communities with the variables analyzed by RDA. Groups are as follows: Fanjing Mountain Nature Reserve Plot A, Plot B, and Plot C. The significance of variables was tested using ANOVA. The variation partitioning analysis was computed using the significant variables identified within each category. Significance levels are * *p* < 0.05.

**Figure 13 microorganisms-12-01061-f013:**
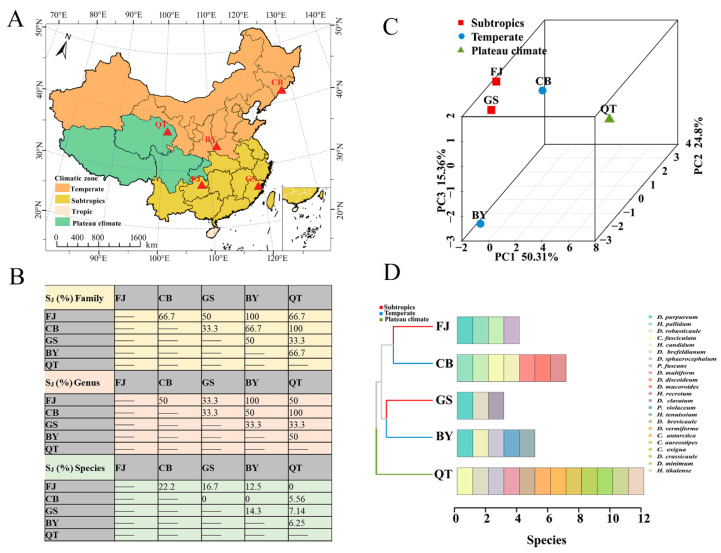
Analysis of dictyostelid diversity comparisons and geographical characteristics in different protected areas. (**A**) Map of the protected area. (**B**) Jaccard similarity coefficients (S_J_ %) of dictyostelids in family, genus, and species within Changbai Mountain in Jilin Province (CB), Gushan Mountain in Fujian Province (GS), Baiyun Mountain in Henan Province (BY), Qinghai–Tibet Plateau in Tibet (QT), and Fanjing Mountain in Guizhou Province (FJ). (**C**,**D**) Beta diversity was analyzed using PCA; bar chart clustered by Bray–Curtis similarities calculated based on the dictyostelid species of the three climate zones.

**Table 1 microorganisms-12-01061-t001:** List of localities sampled in the Fanjing Mountain Nature Reserve in Guizhou Province.

Soil Nos.	Quadrats	Plots	Vegetation	Coordinates	Elevation (m)	Temperature	Air Humidity	pH	Dictyostelid Species
6972	A1	P1	Coniferous forest	27°54′29″ N, 108°41′59″ E	1971	24–37 °C	0.49	4.32	-
6973	A2			27°54′31″ N, 108°41′57″ E	1978	24–37 °C	0.39	5.7	*** Polysphondylium fuscans* (1)
6974	A3			27°54′37″ N, 108°41′53″ E	2049	24–37 °C	0.39	5.85	-
6975	B1	P2	Broadleaf forest	27°50′32″ N, 108°46′30″ E	503	24–37 °C	0.63	4.96	-
6976	B2			27°50′33″ N, 108°46′28″ E	519	24–37 °C	0.6	5.68	-
6977	B3			27°50′32″ N, 108°46′31″ E	656	24–37 °C	0.63	5.65	** Dictyostelium robusticaule* (2)
6978	C1	P3	Mixed forest	27°50′40″ N, 108°47′10″ E	830	24–37 °C	0.6	4.55	-
6979	C2			27°50′39″ N, 108°47′13″ E	842	24–37 °C	0.65	5.61	*D. purpureum* (1), *Heterostelium pallidum* (1)
6980	C3			27°50′41″ N, 108°47′11″ E	818	24–37 °C	0.62	5.83	*Heterostelium pallidum* (29)

The numbers (1), (2), (29) indicate the numbers of species clones obtained from the soil samples (n). * refers to a species new to the Guizhou Province. ** refers to a species new to China.

**Table 2 microorganisms-12-01061-t002:** Morphological characteristics of dictyostelids in the Fanjing Mountain Nature Reserve in Guizhou Province.

Species	*Polysphondylium fuscans* Perrigo and Romeralo (Figure 5)	*Dictyostelium robusticaule* Y. Li, P. Liu, Y. Zou (Figure 6)	*Dictyostelium purpureum* Olive (Figure 7)	*Heterostelium pallidum* (Olive) S. Baldauf, S. Sheikh, and Thulin (Figure 8)
**Characteristic description**	Solitary, irregular (1–3) whorls	Solitary or gregarious, erect or semierect, branches or rarely with one branch	Solitary, strongly phototropic	Gregarious, phototropic, irregular whorls of branches, semierect or prostrate
**Sori**	Dark purple coloration	White	Dark vinaceous purple to almost black	White
**Aggregations**	Radiate	Radiate	Radiate	Initially mound-like, later radiate
**Pseudoplasmodia**	Migrate with stalk	Not migrating	Migrate with stalk	Producing clustered sorogens
**Sorophore tips**	Clavate, two tiers of cells	Obtuse, collar structure, several tiers of cells	Capitate, one or two tiers of cells	Acuminate, one tier of cells
**Sorophore bases**	Round to conical, several tiers of cells	Clavate, several tiers of cells	Robust, basal support disk	Clavate, two or several tiers of cells
**Spores**	Elliptical, inconspicuous polar granules	Oblong or elliptical, without polar granules	Elliptical, without polar granules	Elliptical, inconspicuous polar granules
**Sori diameter (mm)**	0.099–0.157	0.140–0.268	0.147–0.628	0.038–0.140
**Sorocarps (mm)**	2.785–5.473	0.628–2.316	4.101–6.192	2.401–3.047
**Aggregation size (mm)**	1.918–4.015 × 1.794–3.317	5.008–5.731 × 5.683–6.773	3.094–9.461 × 3.135–10.219	3.244–4.562 × 4.276–5.821
**Sorophore (μm)**	13.333–31.037	12.345–40.136	27.321–38.701	9.245–15.515
**Spores (μm)**	3.178–3.755 × 1.316–1.897	6.109–7.754 × 3.414–4.162	5.121–9.081 × 2.330–4.324	6.095–7.041 × 3.269–4.171
**Sorophore tips (μm)**	5.436–6.101	10.443–37.365	4.291–19.829	3.818–4.693
**Sorophore bases (μm)**	7.085–16.743	8.316–19.382	36.464–49.140	16.614–22.085
**Strain no.**	HMJAU A241	HMJAU B341, B321	HMJAU C211	HMJAU C231; C311–C315; C321–C324; C331–C336; C341–C347; C351–C357
**Soil nos.**	6973	6977	6979	6979, 6980
**Collection site**	Fanjing Mountain Nature Reserve (A2)	Fanjing Mountain Nature Reserve (B3)	Fanjing Mountain Nature Reserve (C2)	Fanjing Mountain Nature Reserve (C2), Fanjing Mountain Nature Reserve (C3)
**Vegetation**	Coniferous forest	Broadleaf forest	Mixed forest	Mixed forest
**Distribution in China**	Guizhou	Jilin, Guizhou	Beijing, Taiwan, Fujian, Henan, Guizhou	Taiwan, Jilin, Guizhou
**World distribution**	Sweden, China	China	India, Nepal, Japan, China	Japan, Liberia, Malaysia, Nepal, Philippines, Tanzania, United States, Canada, France, India, Indonesia, Italy, China

## Data Availability

Data are contained within the article and Appendix A.

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
