# Peer review of "Diversity of Cellular Slime Molds (Dictyostelids) in the Fanjing Mountain Nature Reserve and Geographical Distribution Comparisons with Other Representative Nature Reserves in Different Climate Zones of China"

_microorganisms, 2024, doi:10.3390/microorganisms12061061_

Round 1

Reviewer 1 Report

Comments and Suggestions for Authors

The research is well within the scope of the Microorganisms Journal. The literature review justifies the research well and introduces the reader to the topics presented in the manuscript.

The "Introduction" chapter introduces the reader well to the topics described in the manuscript.

The "Materials and Methods" chapter - the authors correctly describe the research methods used in their research. They document their field of research well.

The "Results" chapter - the authors describe the results obtained correctly and sufficiently. The results are well illustrated with graphs and photographs.

"Discussion" chapter - the authors have related their research findings to other studies. This chapter is well written.

The "References" chapter - does not raise any objections. The authors have used all the necessary literature.

 In general, the whole manuscript is correctly written. The only suggestion is to add a "Conclusion" chapter in which they summarise the main findings of their research and write what the strengths and weaknesses of the research are.

Author Response

Response to Reviewer 1 Comments

Dear reviewer:

Thank you for your comments relating to our manuscript [entitled “Diversity of cellular slime molds (dictyostelids) in the Fanjing Mountain Nature Reserve and geographical distribution comparisons with other representative nature reserves in different climate zones of China”] (ID: microorganisms-2994770). The comments were very valuable in addition to being helpful in our efforts to revise and improve our manuscript as well as bringing out the important significant points of our research. We have read the comments carefully and made corrections accordingly. Revised parts are marked in blue in the manuscript. The main corrections in the paper and our responses to the comments are given below. We hope that the revisions in the manuscript and our accompanying responses will be sufficient to make our manuscript suitable for publication in Microorganisms.

Comments and Suggestions for Authors

The research is well within the scope of the Microorganisms Journal. The literature review justifies the research well and introduces the reader to the topics presented in the manuscript.

The “Introduction” chapter introduces the reader well to the topics described in the manuscript.

The “Materials and Methods” chapter - the authors correctly describe the research methods used in their research. They document their field of research well.

The “Results” chapter - the authors describe the results obtained correctly and sufficiently. The results are well illustrated with graphs and photographs.

Discussion” chapter - the authors have related their research findings to other studies. This chapter is well written.

The “References” chapter - does not raise any objections. The authors have used all the necessary literature.

In general, the whole manuscript is correctly written. The only suggestion is to add a “Conclusion” chapter in which they summarise the main findings of their research and write what the strengths and weaknesses of the research are.

Response: The suggestion you’ve mentioned is very meaningful.

We have added the “Conclusion” chapter in the manuscript which summarizes the main findings of research and the strengths and weaknesses of the research.

Modify as follows:

  1. Conclusion

In this study, 34 isolates representing four species of dictyostelids were obtained from soil samples collected from Fanjing Mountain of Guizhou Province, China. One species (Polysphondylium fuscans) was new to China, which provides a new starting point for the study of this species. Our data indicate that the distribution patterns of dictyostelids are most influenced by vegetation and temperature, which provides important information for understanding the ecological characteristics of these organisms. These obvious geographical distribution patterns of dictyostelids in different climate zones has great significance for developing a better understanding of their ecological adaptability and evolutionary history.

However, in future studies, we should (1) consider the limitations of the research area, such as increasing the collection of samples and adopting new technologies such as high-throughput sequencing; (2) consider the complexity of environmental factors, such as soil conditions and the influence of biological factors; and (3) propose specific protection measures or recommendations to help protect these precious biological resources.

We tried our best to improve the manuscript and made some changes marked in blue in revised paper. These will not influence the content and framework of the paper. We appreciate the careful editing by the Editor/Reviewers and earnestly hope the revision will meet with your approval. Once again, thank you very much for your comments and suggestions.

Kind regards,

Pu Liu

E-mail address: [email protected]

Reviewer 2 Report

Comments and Suggestions for Authors

All in the MS

Author Response

Response to Reviewer 2 Comments

Dear reviewer:

Thank you for your comments relating to our manuscript [entitled “Diversity of cellular slime molds (dictyostelids) in the Fanjing Mountain Nature Reserve and geographical distribution comparisons with other representative nature reserves in different climate zones of China”] (ID: microorganisms-2994770). The comments were very valuable in addition to being helpful in our efforts to revise and improve our manuscript as well as bringing out the important significant points of our research. We have read the comments carefully and made corrections accordingly. Revised parts are marked in blue in the manuscript. The main corrections in the paper and our responses to the comments are given below. We hope that the revisions in the manuscript and our accompanying responses will be sufficient to make our manuscript suitable for publication in Microorganisms.

Comments and Suggestions for Authors

  1. Line 16 – “important ecological functions” which are they?

Response: We have added the specific ecological functions.

Modify as follows:

Dictyostelids are unique protists known to have important ecological functions in promoting soil and plant health through their top-down regulation of ecosystem processes, such as decomposition, that involve bacterial populations.

  1. Line 21 – “sampled sites in other protected” how many areas are compared?

Response: We compared the data obtained from Fanjing Mountain Nature Reserve with the other four protected areas.

Modify as follows:

We compared the biodiversity data of dictyostelids in Fanjing Mountain with similar data from previously sampled sites in four other protected areas including Changbai Mountain (CB), Gushan Mountain (GS), Baiyun Mountain (BY) and Qinghai-Tibet Plateau (QT) in China.

  1. Line 29 – “similarity coefficients” which coefficients?

Response: This refers to Jaccard similarity coefficient, also known as Jaccard index, which is used to compare the similarity and difference between finite sample sets. The higher the value of the Jaccard coefficient, the higher the similarity of the samples being compared.

Modify as follows:

and the Jaccard similarity coefficient (Jaccard index) of family, genus and species

  1. Line 33 – “Keywords” order in alphabetic way

Response: We have arranged the keywords in alphabetical order.

Modify as follows:

Keywords: Dictyostelids diversity; Geographical distribution; Protected areas; Soil protists; Taxonomy.

  1. Lines 36-43 – add this in the Abstract?

Response: We have deleted this paragraph.

  1. Line 79 – “ macrocephalum” complete

Response: For a variety of species with the same genus name, one should write the genus name of the first species and abbreviate the genus name thereafter.

Modify as follows:

have yielded only nine species (Dictyostelium macrocephalum, D. implicatum, D. crassicaul, D. firmibasis, D. purpureum, Heterostelium tenuissimum, H. pallidum, Polysphondylium violaceum, and Cavenderia delicata) from Guizhou Province [26,27]

  1. Line 117 – “Table 1”in the Table some words are in bold?

Response: We have carefully corrected the fonts in the table to be consistent.

Modify as follows:

Soil Nos.

Quadrats

Plots

Vegetation

Coordinates

Elevation
(m)

Temperature

Air humidity

pH

Dictyostelid species

6972

A1

P1

Coniferous forest

27°54′29″N,

108°41′59″E

1971

24~37℃

0.49

4.32

——

6973

A2

27°54′31″N, 108°41′57″E

1978

24~37℃

0.39

5.7

**Polysphondylium fuscans (1)

6974

A3

27°54′37″N, 108°41′53″E

2049

24~37℃

0.39

5.85

——

6975

B1

P2

Broadleaf forest

27°50′32″N, 108°46′30″E

503

24~37℃

0.63

4.96

——

6976

B2

27°50′33″N, 108°46′28″E

519

24~37℃

0.6

5.68

——

6977

B3

27°50′32″N, 108°46′31″E

656

24~37℃

0.63

5.65

*Dictyostelium  robusticaule (2)

6978

C1

P3

Mixed forest

27°50′40″N, 108°47′10″E

830

24~37℃

0.6

4.55

——

6979

C2

27°50′39″N, 108°47′13″E

842

24~37℃

0.65

5.61

D. purpureum (1),
Heterostelium  pallidum
(1)

6980

C3

27°50′41″N, 108°47′11″E

818

24~37℃

0.62

5.83

Heterostelium  pallidum (29)

  1. Line 135 – The formulae [29,31] used to, not join

Response: We have added the corresponding reference under each formula, which will enable readers to find the original source more quickly and thus enhance the readability of the article.

Modify as follows:

  1. What is Sj, j=XXX?

Response: SJ is the Jaccard similarity coefficient, j= Jaccard

Modify as follows:

Jaccard similarity coefficient (SJ) (%) [32]

  1. Line 207 – “Jaccard similarity coefficient” include this name in the abstract?

Response: We have added the “Jaccard similarity coefficient” to the abstract.

Modify as follows:

and the Jaccard similarity coefficient (Jaccard index) of family, genus and species

  1. Line 244 – “Table 2” add complete scientific name

Response: We have added the complete scientific name in each instance.

Modify as follows:

Species

Polysphondylium fuscans Perrigo & Romeralo (Fig. 5)

Dictyostelium robusticaule Y. Li, P. Liu, Y. Zou (Fig. 6)

Dictyostelium purpureum Olive

 (Fig. 7)

Heterostelium pallidum (Olive) S. Baldauf, S. Sheikh & Thulin (Fig. 8)

Characteristic
description

Solitary,
irregular (1–3) whorls

Solitary or gregarious,
erect or semi-erect, branches
or rarely with one branch

Solitary,
strongly phototropic

Gregarious, phototropic,
irregular whorls of branches,
semierect or prostrate

Sori

Dark purple coloration

White

Dark vinaceous
purple to almost black

White

Aggregations

Radiate

Radiate

Radiate

Initially mound-like, later radiate

Pseudoplasmodia

Migrate with stalk

Not migrating

Migrate with stalk

Producing clustered sorogens

Sorophore tips

Clavate,
two tiers of cells

obtuse, collar structure,
several tiers of cells

Capitate,
one or two tier of cells

Acuminate,
one tier of cells

Sorophore bases

Round to conical,
several tiers of cells

Clavate,
several tiers of cells

Robust,
basal support disk

Clavate,
two or several tiers of cells

Spores

Elliptical,
inconspicuous polar granules

Oblong or elliptical,
without polar granules

Elliptical,
without polar granules

Elliptical,
inconspicuous polar granules

Sori diameter (mm)

0.099–0.157

0.140–0.268

0.147–0.628

0.038–0.140

Sorocarps (mm)

2.785–5.473

0.628–2.316

4.101–6.192

2.401–3.047

Aggregations size (mm)

1.918–4.015×1.794–3.317

5.008–5.731×5.683–6.773

3.094–9.461×3.135–10.219

3.244–4.562×4.276–5.821

Sorophore (μm)

13.333–31.037

12.345–40.136

27.321–38.701

9.245–15.515

Spores(μm)

3.178–3.755×1.316–1.897

6.109–7.754×3.414–4.162

5.121–9.081×2.330–4.324

6.095–7.041×3.269–4.171

Sorophore tips(μm)

5.436–6.101

10.443–37.365

4.291–19.829

3.818–4.693

Sorophore bases(μm)

7.085–16.743

8.316–19.382

36.464–49.140

16.614–22.085

Strain No.

HMJAU A241

HMJAU B341, B321

HMJAU C211

HMJAU C231; C311-C315; C321- C324; C331-C336; C341-C347; C351-C357

Soil Nos.

6973

6977

6979

6979, 6980

Collection site

Fanjing Mountain
Nature Reserve (A2)

Fanjing Mountain
Nature Reserve (B3)

Fanjing Mountain
Nature Reserve (C2)

Fanjing Mountain Nature Reserve (C2),
Fanjing Mountain Nature Reserve (C3)

Vegetation

Coniferous forest

Broadleaf forest

Mixed forest

Mixed forest

Distribution in China

Guizhou

Jilin, Guizhou

Beijing, Taiwan, Fujian,
Henan, Guizhou

Taiwan, Jilin, Guizhou

World distribution

Sweden, China

China

India, Nepal,
Japan, China

Japan, Liberia, Malaysia, Nepal, Philippines, Tanzania, United States, Canada, France, India, Indonesia, Italy, China,

  1. Line 246 – “ fuscans” scientific name complete

Response: We have added the complete scientific name.

Modify as follows:

Morphological features of Polysphondylium fuscans Perrigo & Romeralo

  1. Line 246 – “ robusticaule” scientific name complete

Response: We have added the complete scientific name.

Modify as follows:

Morphological features of Dictyostelium robusticaule Y. Li, P. Liu, Y. Zou

  1. Line 246 – “ purpureum” scientific name complete

Response: We have added the complete scientific name.

Modify as follows:

Morphological features of Dictyostelium purpureum Olive

  1. Line 246 – “ pallidum” scientific name complete

Response: We have added the complete scientific name.

Modify as follows:

Morphological features of Heterostelium pallidum (Olive) S. Baldauf, S. Sheikh & Thulin

We have tried our best to improve the manuscript and have made some changes that marked in blue in revised paper. This do not influence the content and framework of the paper. We appreciate the efforts by the Editor/Reviewers to improve the manuscript and hope the revision will meet with your approval. Once again, thank you very much for your comments and suggestions.

Kind regards,

Pu Liu

E-mail address: [email protected]